# Gender Influences Virtual Reality-Based Recovery of Cognitive Functions in Patients with Traumatic Brain Injury: A Secondary Analysis of a Randomized Clinical Trial

**DOI:** 10.3390/brainsci12040491

**Published:** 2022-04-12

**Authors:** Roberta Bruschetta, Maria Grazia Maggio, Antonino Naro, Irene Ciancarelli, Giovanni Morone, Francesco Arcuri, Paolo Tonin, Gennaro Tartarisco, Giovanni Pioggia, Antonio Cerasa, Rocco Salvatore Calabrò

**Affiliations:** 1Institute for Biomedical Research and Innovation (IRIB), National Research Council of Italy, 98164 Messina, Italy; roberta.bruschetta@irib.cnr.it (R.B.); gennaro.tartarisco@irib.cnr.it (G.T.); giovanni.pioggia@irib.cnr.it (G.P.); 2Department of Engineering, Università Campus Bio-Medico di Roma, Via Alvaro del Portillo 21, 00128 Rome, Italy; 3Department of Biomedical and Biotechnological Science, University of Catania, 95123 Catania, Italy; mariagraziamay@gmail.com; 4Stroke Unit, Azienda Ospedaliera Universitaria Gaetano Martino, 98123 Messina, Italy; g.naro11@alice.it; 5Department of Life, Health and Environmental Sciences, University of L’Aquila, 67100 L’Aquila, Italy; irene.ciancarelli@univaq.it (I.C.); g.morone@hsantalucia.it (G.M.); 6S’Anna Institute, 88900 Crotone, Italy; f.arcuri@isakr.it (F.A.); patonin18@gmail.com (P.T.); 7Pharmacotechnology Documentation and Transfer Unit, Preclinical and Translational Pharmacology, Department of Pharmacy, Health Science and Nutrition, University of Calabria, 87036 Rende, Italy; 8IRCCS Centro Neurolesi “Bonino Pulejo”, 98124 Messina, Italy; salbro77@tiscali.it

**Keywords:** traumatic brain injury, virtual reality, neurorehabilitation, cognitive recovery, gender

## Abstract

The rehabilitation of cognitive deficits in individuals with traumatic brain injury is essential for promoting patients’ recovery and autonomy. Virtual reality (VR) training is a powerful tool for reaching this target, although the effectiveness of this intervention could be interfered with by several factors. In this study, we evaluated if demographical and clinical variables could be related to the recovery of cognitive function in TBI patients after a well-validated VR training. One hundred patients with TBI were enrolled in this study and equally randomized into the Traditional Cognitive Rehabilitation Group (TCRG: n = 50) or Virtual Reality Training Group (VRTG: n = 50). The VRTG underwent a VRT with BTs-N, whereas the TCRG received standard cognitive treatment. All the patients were evaluated by a complete neuropsychological battery before (T0) and after the end of the training (T1). We found that the VR-related improvement in mood, as well as cognitive flexibility, and selective attention were influenced by gender. Indeed, females who underwent VR training were those showing better cognitive recovery. This study highlights the importance of evaluating gender effects in planning cognitive rehabilitation programs. The inclusion of different repetitions and modalities of VR training should be considered for TBI male patients.

## 1. Introduction

Virtual rehabilitation (VR) [1], as an emerging technology, is an opportunity in health care and a new tool for this specific purpose [2], being able to carry out the evaluation and treatment of cognitive and behavioral disorders while simulating a real environment. In these last decades, many attempts have been made to evaluate and train attention, memory, executive function, and spatial perception through VR programs in patients with stroke [3]. The specific VR characteristics (flexibility, sense of presence, and emotional engagement) has allowed researchers to obtain good results in motor and cognitive rehabilitation, as well as in patients with TBI, although the data concerning the effects on cognitive function and quality of life are more limited than the motor ones [3,4].

Generally, several studies have reported the successful intervention of VR application after TBI without adverse effects [5,6] and with clinical improvement in memory [7,8,9,10], executive function [11,12,13], and attention [7,14]. However, although VR could be a useful instrument both in the assessment and in the rehabilitation treatment of TBI, some considerations must be reported. Wright et al. [15] suggest implementing the VR assessment protocols for mild TBI because of the subtle deficits that are hard to detect with traditional instruments, while the VR treatment protocols seem to be equally effective for mild to severe TBI [16]. Two principal limitations restrict the use of VR in clinical practice: accessibility and cost [8]. However, since the long-term deficits in TBI concern cognition and behavior [17], it is necessary to highlight that VR may contribute to reducing these kinds of deficits in these individuals, as demonstrated by VR interventions on memory, attention, executive function, behavioral control, and regulation of mood [2]. Indeed, VR, thanks to its multisensory approach, can stimulate and enhance the spontaneous post-TBI regeneration processes, which otherwise may be too weak to counter the deterioration of the TBI-damaged brain areas, including the frontal lobes. The exercises performed in a virtual environment help the patient to develop the knowledge of the results of the movements (knowledge of the results) and the knowledge of the quality of the movements (knowledge of the performances), which positively affect a patient’s functional recovery, including the cognitive one. Thus, VR could allow for greater results than conventional training, through global stimulation and dual cognitive and motor tasking, which allow for greater patient involvement [4].

Another interesting aspect, although neglected, is the evaluation of possible demographic factors that could mediate patterns of recovery due to VR intervention. For instance, gender has been proposed as one of the main factors influencing the cognitive changes after rehabilitation treatment in patients with brain injury. Most of the evidence has been reported in stroke patients, although with conflicting results [18,19,20]. In TBI patients, a gender-related effect on cognitive outcomes following advanced neurorehabilitation remains to be demonstrated. Indeed, a recent systematic review has shown that genetic disposition, baseline cognitive status, severity and frequency of injury, age, and sex were predictors of cognitive performance across time [21]. Moreover, it has been suggested that females may have better perceived cognitive functional outcomes than males one year after severe TBI, probably due to different hormone levels [22]. 

In a previous study, we have demonstrated that patients with TBI receiving both conventional and VR training achieved significant improvements in different cognitive and mood domains, although VR led to better results. Moreover, only TBI patients in the VR group improved in specific cognitive domains, such as cognitive flexibility, attentional shifting, visual search, and executive and visuospatial functions, which are necessary for planning and managing daily life. 

The aim of this study was to assess the factors associated with better functional recovery in patients with TBI receiving either conventional or VR cognitive rehabilitation. In particular, we sought to investigate whether and to what extent a gender effect in determining cognitive outcomes in such patients may exist. 

## 2. Materials and Methods

### 2.1. Study Population

This study was a secondary analysis of data from a prospective randomized controlled trial [23] that examined the effects of VR treatment on cognitive outcomes in 100 patients with TBI. Details of the main study are provided elsewhere [23], but, in brief, one hundred patients with a TBI, who attended the Behavioral Neurorehabilitation and Robotics Unit of The IRCCS Centro Neurolesi “Bonino Pulejo” (Messina, Italy) from January 2016 to December 2018, were enrolled in the study and randomized into the Traditional Rehabilitation Control Group (TRCG: n = 50) or in the Virtual Reality Training Group (VRTG: n = 50) (Table 1; for more information, see ref [23]).

Inclusion criteria were (i) neurologic diagnosis of mild to moderate TBI in the post-acute phase (i.e., 3 to 6 months after the acute event) and (ii) presence of mild to moderate cognitive impairment (as per the Montreal Cognitive Assessment (MoCA) score: 18 to 25). The exclusion criteria were (i) age > 85 years; (ii) presence of disabling sensory abnormalities, including visual and hearing loss; (iii) uncontrolled epilepsy, with seizures having positive symptoms, such as audiovisual hallucinations; and (iii) medical (i.e., heart and pulmonary failure) and psychiatric disorders (severe depression and anxiety, and/or psychosis) potentially interfering with the VR training.

The present study was conducted in accordance with the 1964 Helsinki Declaration and approved by our Research Institute Ethics Committee (ID 25/2015); written informed consent was obtained from all participants.

### 2.2. Study Design and Treatment

The experimental procedure of this study has previously been reported [23]. In brief, using a double-blind randomized controlled design, TBI patients underwent the same amount of cognitive rehabilitation treatment (24, 1-h sessions, 3 times per week for 8 weeks), but using different tools. Individuals in the TRCG group underwent traditional cognitive rehabilitation, administered in individual sessions using a face-to-face interaction between therapist and patient with paper and pencil activities, whereas VRTG performed a VRT using the BTs-Nirvana (BTs-N) system (https://www.btsbioengineering.com/nirvana/, accessed on 1 June 2016). The BTs-N program was based on interactive sessions through virtual scenarios in which the patient could carry out training with the help of a therapist. The patient interacted with 5 virtual scenarios for each session, which provided audiovisual stimuli through the patient’s movement, creating total sensory involvement (Figure 1). The schemes allowed users to select some virtual elements that remained visible for a variable time, and to be able to move or view them in the correct position.

Both treatments targeted activities that increased attention, visual-spatial, and executive skills. In particular, the TRCG group underwent exercises in a specific physical space (rehabilitation table) for stimulating simple associations (i.e., letter-color), inhibitory control, divided attention (targeting stimuli in relation to specific visual and semantic characteristics neglecting the distractors), arithmetic operations estimating the numerical quantity and the categorization, and deductive logical reasoning. Otherwise, the patients enrolled in the VRTG group performed the same cognitive task in a virtual environment through the movement performed on the interactive screen (Figure 1). The movements allow the user to manipulate specific objects in different directions (i.e., balls, flowers, and butterflies), create specific associations (i.e., color-number), or explore some elements (colors, musical arcs, geometric shapes, animals, etc.) with a dynamic interaction in the virtual environment. The difficulty level increases with the increment of the complexity of the task, elements on the screen (or table), and greater difficulty of the requests by the therapist.

Moreover, both groups also underwent the same conventional physiotherapy program carried out by expert therapists consisting of occupational therapy exercises together with passive/active mobilization of upper and lower limbs, trunk control, standing, and ambulation. Physiotherapists and psychologists, as well as data entry assistants, were blinded to all phases of the study.

### 2.3. Outcome Measures

Each participant was assessed by means of a neuropsychological evaluation before (T0) and immediately after the end of the training (T1). The neuropsychological battery included: (a) Montreal Cognitive Assessment (MoCA) to assess the general cognitive state; (b) Hamilton Rating Scale Depression (HRS-D) and Hamilton Rating Scale Anxiety (HRS-A) to assess mood and anxiety, respectively; (c) Frontal Assessment Battery (FAB) and Weigl’s Test to evaluate frontal abilities; and (d) Visual Search (VS) and Trial Making Test (TMT) to measure the attention process, attentive shifting, and visual research abilities. 

### 2.4. Statistical Analysis

Statistical analyses were performed using the computer software RStudio Version 4.0.3 (RStudio, Boston, MA, USA; https://cran.r-project.org, accessed on 10 October 2020), considering *p* < 0.05 as statistically significant.

To assess the influence of group (TCRG vs. VRTG), demographic variables (Gender, Age, Marital Status, number of children, Educational Level, Disease Duration), and Functional Independence Measure (FIM) and MoCA at admission on the improvement of the neuropsychological performance, we employed eight different linear regression models using, as the outcome, the difference of each parameter between time 0 and time 1. Behavioral changes associated with cognitive treatment were calculated as differential T1 − T0 (delta) scores. 

## 3. Results

All of the patients completed the training program without any adverse events, including cyber-sickness. As shown in the previous work (Table 1, [23]), no significant differences were found for every demographic and clinical variable between the two groups, and VRTG treatment induced a significant improvement in all the cognitive and emotional neuropsychological scales.

The linear regression model confirmed that the effect of treatment was the only factor associated with cognitive improvement in all cognitive domains (TCRG vs. VRTG). However, recovery of cognitive and emotional functions was also influenced by other factors. As shown in Table 2, four specific outcome measures demonstrated an additional factor influencing recovery. Considering anxiety level (HRS-A), a significant Group × Gender interaction effect was detected before and after treatment (F = 5.93; *p* = 0.017). Figure 2 shows that TBI females are characterized by a lower reduction of anxiety levels after VRTG, with respect to males. As for VS and WEIGL scores, an additional significant effect of gender was also detected, where females showed the highest performance (*p* = 0.016, r2 = 0.419, and beta-value = 0.44836; *p* = 0.022, r2 = 0.528, and beta-value = 0.3819, respectively, for VS and WEIGL). Finally, the better performance in the TMT-B-A after VR treatment was also related to the FIM scores at admission (*p* = 0.013; r2 = 0.273; beta-value = −0.25810).

## 4. Discussion

In our previous analysis [23], we concluded that VRTG has a significant impact on the cognitive recovery of TBI patients, thanks to the multisensory approach embedded into the VR system that can stimulate and enhance the spontaneous post-TBI regeneration processes. The exercises performed in a virtual environment help to develop the knowledge about the results and the quality of the movements (biofeedback) that positively affect patient’s functional recovery, including the cognitive one [24]. 

In this new study, we demonstrated that part of this effect is influenced by gender and other clinical factors, including anxiety level and FIM scores. 

In particular, we found a gender-dependent effect in some important domains: attention, frontal abilities, and anxiety. Generally, there is a lack of understanding about trauma-induced changes in females and males. Many clinical and pre-clinical trials have been focused on males [25], although it has been demonstrated that in the elderly population, TBI is more prevalent in females [26]. Evidence from animal studies suggests that progesterone and estrogen may offer protective effects in secondary brain injury [27]. Wright et al. [28] found that administration of progesterone after severe TBI was associated with a lower mortality rate at 30 days post injury. Due to varying sex hormone levels, which exert their effects on cerebral organization and neuroplasticity, gender differences might also influence patient outcomes [29]. In particular, a recent review has demonstrated a pivotal role for neuron-derived 17β-estradiol (E2) in the regulation of synaptic plasticity, memory, socio-sexual behavior, and sexual differentiation, as well as injury-induced reactive gliosis and neuroprotection [30]. Furthermore, growing evidence suggests that astrocyte-derived E2, which is induced following brain injury, plays a key role in reactive gliosis, neuroprotection, and cognitive preservation. This could represent the neurobiological basis for the better cognitive outcomes that we found in females as compared to males.

Several studies found that females were more likely to be diagnosed with depression and/or anxiety following TBI [31]. In our study, we found a Group*Gender interaction effect on anxiety scores. This finding is in line with evidence provided by Yue et al. [32], who proposed that gender may interact with age for post-traumatic stress disorders, with young TBI females at high risk. 

Gender may also be a potential moderator of the effects of exercise on cognitive function, given that the female sex may positively influence the strength of the relationship between exercise and cognition [33]. An early systematic review reported that studies involving a high proportion of healthy older females demonstrated greater improvements in cognition following exercise training compared with studies involving a high proportion of healthy older males [33]. In line with this evidence, we found that females performed better in attention and executive functions. This additional finding was like those reported by Khattab et al. [34], who found that females with brain injury demonstrated improvements in selective attention and conflict resolution following aerobic exercises.

Finally, another interesting finding is the impact of FIM scores at admission on the performance of TMT. FIM is a scale commonly used in the rehabilitation setting to assess measures of independence for self-care, including sphincter control, transfers, locomotion, communication, and social cognition. For this reason, it is possible that reduced functional status at admission may impact the TMT performance. However, this additional relationship should be considered for further studies.

### Limitations

The main limitations of the study are the lack of a long-term follow-up and the absence of a control group without a cognitive treatment, which could be useful to exclude the variability related to spontaneous recovery independent of the interventions. Finally, our VR treatment has been considered feasible and accepted by all patients although evidence by satisfaction questionnaires was not provided.

## 5. Conclusions

For the first time ever, we found that there was a gender-dependent effect following cognitive rehabilitation using VR, with a significant improvement in some important domains, including attention, frontal abilities, and anxiety. The mechanisms underlying gender differences in the effects of VR treatment on cognitive and emotional recovery of TBI patients are an unexplored field of study, although they may partly depend on neuro-derived female sexual hormones. Thus, gender should be considered before planning rehabilitation programs for TBI patients. Further larger multicenter studies with short and long-term follow up periods and assessing sexual hormone levels, as well as other neuropeptides and growth factors involved in brain plasticity, are needed to determine whether hormones and other molecules influence the cognitive outcomes of traumatic head injury and the trajectory of these outcomes [22].

## Figures and Tables

**Figure 1 brainsci-12-00491-f001:**
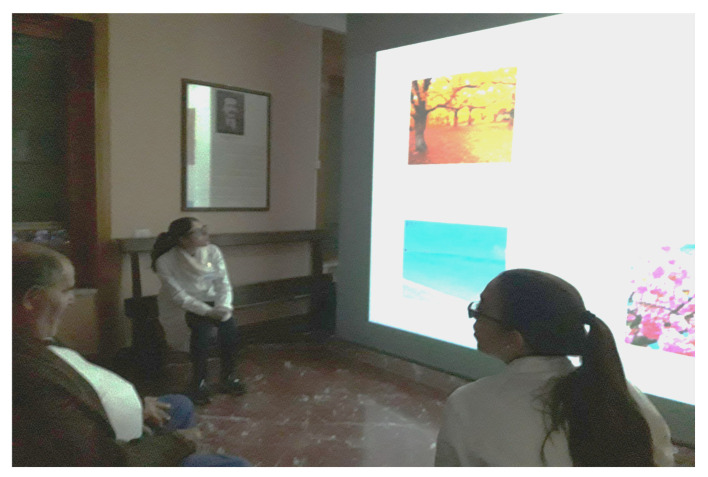
A patient undergoing VR training. The patient must recognize the seasons and operate with images, which emit sounds like the season in which he touches them (jingle bells for the winter). It is also asked to list the activities to be carried out in the various seasons and any recipes to be prepared with seasonal fruit (training for fluency and executive skills).

**Figure 2 brainsci-12-00491-f002:**
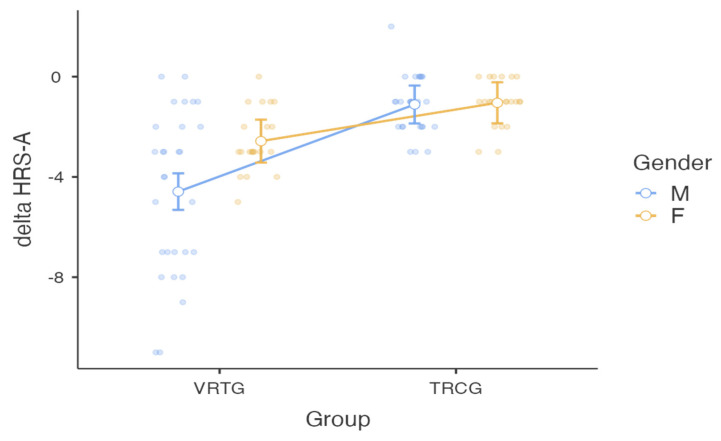
Interaction effect Group × Gender for the HRS-A scores. Behavioral changes before and after treatment were calculated as the delta-value. Traditional Rehabilitation Control Group: TRGC; Virtual Reality Training Group: VRTG.

**Table 1 brainsci-12-00491-t001:** Demographics characteristics at baseline for both groups. Quantitative variables were expressed as means ± standard deviations, and categorical variables as frequencies and percentages.

	Experimental	Control	All
**Participants**	50	50	100
**Age**	38.7 ± 9.3	41.1 ± 10.8	39.9 ± 10.1
**Education**	2.9 ± 0.8	2.7 ± 0.8	2.8 ± 0.850
**Gender**			
*Male*	29 (57.9%)	26 (52%)	56 (56%)
*Female*	21 (42.1%)	24 (48%)	44 (44%)
**Interval from TBI**			
*Mean in months*	5 ± 1	5 ± 1	5 ± 1
**Brain lesion site/side**			
*Cortical right*	22	24	46
*Subcortical right*	16	17	33
*Cortical left*	8	6	14
*Subcortical left*	4	3	7

**Table 2 brainsci-12-00491-t002:** Summary of regression analyses. Hamilton Rating Scale Anxiety (HRS-A); Trial Making Test (TMT B-A); Visual Research (VS).

	HRS-A			TMT-BA			VS			WEIGL		
	Sum of Squares	Mean Square	F/*p-Value*	Sum of Squares	Mean Square	F/*p-Value*	Sum of Squares	Mean Square	F/*p-Value*	Sum of Squares	Mean Square	F/*p-Value*
Gender	16.95	16.9	** 4.2/*0.04* **	46.1	46.1	0.06/*0.79*	153.5172	153.5172	** 6.1/*0.01* **	12.55	12.55	** 5.41/*0.02* **
Group	163.9	163.9	** 40.8/*<0.001* **	12,379	12,379	** 18.1/*<0.001* **	1169.23	1169.23	** 46.2/*<0.001* **	170.96	170.96	** 73.747/*<0.001* **
Age	15.61	15.61	3.8/*0.052*	681.5	681.5	1.1/*0.32*	0.7484	0.7484	0.02/*0.86*	1.17	1.17	0.5/0.48
Educational Level	2.57	0.85	0.2/*0.88*	165.8	55.3	0.08/*0.97*	8.5168	2.8389	0.11/*0.95*	5.89	1.96	0.84/*0.47*
Marital Status	25.61	12.81	3.2/*0.056*	801.2	400.6	0.58/*0.55*	110.1476	55.0738	2.1/*0.12*	6.14	3.07	1.3/*0.27*
N°Children	5.52	1.84	0.4/*0.7*	87.7	29.2	0.04/*0.98*	70.1535	23.3845	0.92/*0.43*	8.81	2.94	1.2/*0.29*
Disease Duration	10.2	10.2	2.5/*0.11*	40.4	40.4	0.05/*0.81*	15.2589	15.2589	0.6/*0.43*	7.69	7.69	3.3/*0.07*
Moca at T0	1.1	1.09	0.2/0.603	679.1	679.1	0.99/*0.32*	0.1042	0.1042	0.004/*0.95*	2.09	2.09	0.91/*0.34*
FIM at T0	2.49	2.4	0.6/*0.433*	4361.3	4361.3	** 6.4/*0.01* **	0.0122	0.0122	0.0004/*0.98*	1.12	1.12	0.48/*0.49*
Residuals	341.2	4.01	-	57,823.6	680.3	-	2150.0076	25.2942	-	197.05	2.32	-

## Data Availability

The data presented in this study are available on request from the corresponding author.

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
