# Peer review of "Gender Influences Virtual Reality-Based Recovery of Cognitive Functions in Patients with Traumatic Brain Injury: A Secondary Analysis of a Randomized Clinical Trial"

_brainsci, 2022, doi:10.3390/brainsci12040491_

Round 1

Reviewer 1 Report

- The paper describes a study to investigate the relation of demographical (especially Gender) and clinical variables  with the recovery of cognitive functions on TBI patients after a VR training. The study used the data from a study made previously and described in another article. A long term follow-up is needed to derive more precise conclusions.

- The paper reports on a relevant topic, with high social impact since it aims to improve rehabilitation programs in TBI patients.

- The paper is well written, structured and organized.
- The methodology used is clearly described and with adequate information
- In discussion authors relate and compare the results obtained with other studies but relation with the obtained data could be more clearly explained.
- Discussion and conclusions are adequate and reflect the major findings.
- Main limitations of the study are identified and described.

The paper should be improved by the following:

- the references in the text of all the article should be in square brackets instead of rounded brackets.

- in line 44, introduce a white space before the word "flexibility"

- In line 100, section 2.2, the article misses the definition of "CR". Authors should mention in the article what "CR" stands for.

- in line 104, authors should refer what "BTs-N" stands for.

- in line 130, is missing a decimal point in "< 0 05"

- In line 132, section 2.4: authors should define the acronym "FIM" in its first use.

- In line 140, section 3: has a reference to "Figure 1" but it should be to "Figure 2"

- in line 153, section 3: there are white spaces between the words "performance" and "detected" that should be removed.

Author Response

REVIEWER N°1

The paper describes a study to investigate the relation of demographical (especially Gender) and clinical variables  with the recovery of cognitive functions on TBI patients after a VR training. The study used the data from a study made previously and described in another article. A long term follow-up is needed to derive more precise conclusions. The paper reports on a relevant topic, with high social impact since it aims to improve rehabilitation programs in TBI patients. The paper is well written, structured and organized. The methodology used is clearly described and with adequate information
In discussion authors relate and compare the results obtained with other studies but relation with the obtained data could be more clearly explained.
Discussion and conclusions are adequate and reflect the major findings.

REPLY: We would like to express our appreciation for the reviewer’s comments and consideration.

- Main limitations of the study are identified and described. The paper should be improved by the following:

1)  the references in the text of all the article should be in square brackets instead of rounded brackets.

REPLY: Done

2) in line 44, introduce a white space before the word "flexibility"

REPLY: Done

3) In line 100, section 2.2, the article misses the definition of "CR". Authors should mention in the article what "CR" stands for.

REPLY: Done

4) in line 104, authors should refer what "BTs-N" stands for.

REPLY: Done

5)  in line 130, is missing a decimal point in "< 0 05"

REPLY: Done

6) In line 132, section 2.4: authors should define the acronym "FIM" in its first use.

REPLY: Done

7) In line 140, section 3: has a reference to "Figure 1" but it should be to "Figure 2"

REPLY: Done

8)  in line 153, section 3: there are white spaces between the words "performance" and "detected" that should be removed.

REPLY: Done

Reviewer 2 Report

The paper is interesting and appropriate for the Journal. However, I feel that some issues should be addressed in a suitable revision. My points are listed below.

  1. The Introduction should be improved. A more solid and comprehensive theoretical framework should on the specific topic be provided with a strong rationale for the current investigation.
  2. I wonder whether the authors considered the age next to the gender as a predictive factor. Based on my reading, clarification is needed on that point. 
  3. The authors claimed for an interaction between gender and other clinical factors in the Discussion section. The clinical factors acknowledged should be clarified. 
  4. The Conclusion should be enhanced. Future research perspectives should be added. 
  5. English editing is required throughout. 

Author Response

REVIEWER N°2

The paper is interesting and appropriate for the Journal. However, I feel that some issues should be addressed in a suitable revision. My points are listed below.

  1. The Introduction should be improved. A more solid and comprehensive theoretical framework should on the specific topic be provided with a strong rationale for the current investigation.

REPLY: Following the reviewer’s suggestions the Introduction has been modified.

2. I wonder whether the authors considered the age next to the gender as a predictive factor. Based on my reading, clarification is needed on that point.

REPLY: As reported in paragraph 2.4 “Statistical Analysis”, “age” has been included, together with other demographic variables, in our model. Results showed that Age has no significant effect on outcome measures. In the manuscript, the description of the results is focused on significative predictors, but to be thorough, we report here the results of further analyses on the interaction effect between Gender and Age (see attached document)

ANOVA Omnibus tests - Delta HRS-D:

SS

df

F

p

η²

Model

14.6493

3

0.65926

0.579

0.020

Gender

12.5182

1

1.69008

0.197

0.017

Age

0.0158

1

0.00214

0.963

0.000

Gender ✻ Age

1.6295

1

0.21999

0.640

0.002

Residuals

711.0607

96

Total

725.7100

99

ANOVA Omnibus tests - Delta HRS-A:

SS

df

F

p

η²

Model

32.627

3

1.8331

0.146

0.054

Gender

29.334

1

4.9443

0.029

0.049

Age

0.535

1

0.0902

0.765

0.001

Gender ✻ Age

0.227

1

0.0382

0.845

0.000

Residuals

569.563

96

Total

602.190

99

ANOVA Omnibus tests - Delta TMT-A:

SS

df

F

p

η²

Model

79.45

3

0.2179

0.884

0.007

Gender

3.70

1

0.0304

0.862

0.000

Age

5.86

1

0.0482

0.827

0.000

Gender ✻ Age

74.28

1

0.6113

0.436

0.006

Residuals

11665.77

96

Total

11745.22

99

ANOVA Omnibus tests - Delta TMT-B:

SS

df

F

p

η²

Model

2231.1

3

0.6501

0.585

0.020

Gender

12.5

1

0.0109

0.917

0.000

Age

14.8

1

0.0129

0.910

0.000

Gender ✻ Age

2166.7

1

1.8939

0.172

0.019

Residuals

109829.6

96

Total

112060.7

99

ANOVA Omnibus tests - Delta TMT-B-A:

SS

df

F

p

η²

Model

822.17

3

0.33401

0.801

0.010

Gender

351.36

1

0.42823

0.514

0.004

Age

6.82

1

0.00831

0.928

0.000

Gender ✻ Age

442.04

1

0.53875

0.465

0.006

Residuals

78767.43

96

Total

79589.60

99

ANOVA Omnibus tests - Delta MA:

SS

df

F

p

η²

Model

158.83

3

1.4342

0.238

0.043

Gender

157.30

1

4.2614

0.042

0.042

Age

8.91

1

0.2415

0.624

0.002

Gender ✻ Age

2.08

1

0.0564

0.813

0.001

Residuals

3543.73

96

Total

3702.56

99

ANOVA Omnibus tests - Delta WEIGL:

SS

df

F

p

η²

Model

13.3474

3

1.0576

0.371

0.032

Gender

8.1565

1

1.9388

0.167

0.020

Age

0.0476

1

0.0113

0.916

0.000

Gender ✻ Age

4.8294

1

1.1480

0.287

0.012

Residuals

403.8651

96

Total

417.2125

99

In addition, from Table I, it’s possible to observe that for the enrolled population, age ranged from 20 to 55 years. A possible explanation could be related to the lack of elderly subjects in our sample. Indeed, it has been demonstrated that over-65 age people are characterized by a lower amount of improvement after rehabilitation. (https://www.ncbi.nlm.nih.gov/pmc/articles/PMC2600417)

3. The authors claimed an interaction between gender and other clinical factors in the Discussion section. The clinical factors acknowledged should be clarified.

REPLY: Done

4. The Conclusion should be enhanced. Future research perspectives should be added.

REPLY: Following the reviewer’s suggestions the Conclusion has been modified.

5. English editing is required throughout.

REPLY: English editing has been performed throughout the paper.

Reviewer 3 Report

  1. You should better define the intervention of the control and experimental group. What kind of tasks do they perform? What are the therapeutic objectives of both the intervention of the control group and the experimental group? Define it.
  2. In the methodology, ¿are the evaluators blinded? ¿are they the same therapists?, if yes, describe it, and if not, you can include it in the limitations of the study.
  3. Can you describe what the conventional physiotherapy programme consists of?
  4. ¿What functional impact does or can this cognitive improvement have or can have? You should describe it in the discussion.
  5. ¿Have you collected information about patient satisfaction with the use of VR? If yes, please describe it, and if not, you can include it in the limitations of the study.
  6. Have you taken into account possible adverse effects of the use of virtual reality? If yes, please describe, and if no, you can include it in the limitations of the study.

Author Response

  1. You should better define the intervention of the control and experimental group. What kind of tasks do they perform? What are the therapeutic objectives of both the intervention of the control group and the experimental group? Define it.

REPLY: Following the reviewer's suggestion the 2.2 section "Study Design and Treatment" has been completely re-formulated including this additional information. ì

2. In the methodology, ¿are the evaluators blinded? ¿are they the same therapists?, if yes, describe it, and if not, you can include it in the limitations of the study.

REPLY: This is a double-blind randomized controlled trial. This additional information has been included in the 2.2 section.

3. Can you describe what the conventional physiotherapy programme consists of?

REPLY: Done. see section 2.2 in the final part.

4. What functional impact does or can this cognitive improvement have or can have? You should describe it in the discussion.

REPLY: Following the reviewer's suggestion we included a new statement in the first part of the discussion.

5. Have you collected information about patient satisfaction with the use of VR? If yes, please describe it, and if not, you can include it in the limitations of the study.

REPLY: We would like to thank this reviewer for this important suggestion. We now included a new statement in the limitation section about this issue.

Have you taken into account possible adverse effects of the use of virtual reality? If yes, please describe, and if no, you can include it in the limitations of the study.

REPLY: This information has already been reported in the Results section "All of the patients completed the training program without any adverse events, including cyber-sickness. "